# Cleaving Expectations: A Review of Proteasome Functional and Catalytic Diversity

**DOI:** 10.3390/biom15111524

**Published:** 2025-10-29

**Authors:** Daniel Zachor-Movshovitz, Yegor Leushkin, Katharina I. Zittlau, Gili Ben-Nissan, Michal Sharon

**Affiliations:** Department of Biomolecular Sciences, Weizmann Institute of Science, Rehovot 7610001, Israel; daniel.zachor@weizmann.ac.il (D.Z.-M.); yegor.leushkin@weizmann.ac.il (Y.L.); katharina.zittlau@weizmann.ac.il (K.I.Z.); gili.ben-nissan@weizmann.ac.il (G.B.-N.)

**Keywords:** proteasome, protein degradation, catalytic sites, protein substrates

## Abstract

The proteasome is a central proteolytic complex that maintains protein homeostasis by eliminating damaged, misfolded, and regulatory proteins. Beyond this quality control role, it generates bioactive peptides that contribute to immune surveillance, intracellular signaling, neuronal communication, and antimicrobial defense. Proteolysis is mediated by the catalytic β1, β2, and β5 subunits, traditionally defined by caspase-like, trypsin-like, and chymotrypsin-like activities. However, these sites display overlapping and flexible specificities, enabling cleavage after nearly all amino acids. This review focuses on proteasome catalytic activity, with particular emphasis on the biochemical and structural features of the catalytic subunits that define cleavage selectivity. We first provide a historical overview of the discovery of proteolytic activities and trace the evolutionary diversification of subunits that gave rise to specialized variants such as the immunoproteasome, thymoproteasome, intermediate proteasomes, and the spermatoproteasome. We then highlight how advances in computational modeling and structural biology have refined our understanding of cleavage preferences. In addition, we examine how regulatory particles, post-translational modifications, and physiological conditions, including inflammation, oxidative stress, and aging, modulate proteolytic activity. Finally, we discuss the development of selective inhibitors targeting individual catalytic sites, emphasizing their therapeutic potential in cancer, autoimmunity, and infectious disease, and outline future directions for the field.

## 1. Introduction

The proteasome is a large, multi-subunit protease complex that plays a central role in maintaining protein homeostasis [1]. It functions primarily to degrade intracellular proteins, thereby eliminating damaged or misfolded proteins that could otherwise accumulate and disrupt cellular functions [2]. In addition to this quality control role, the proteasome also removes short-lived regulatory proteins that have completed their function, ensuring precise regulation of cellular processes such as the cell cycle, transcription, and signal transduction [2]. Through the targeted removal of specific proteins, the proteasome enables dynamic remodeling of the cellular proteome in response to environmental cues and internal signals [3]. Most peptides generated during this proteolytic process are further hydrolyzed into amino acids, which are recycled to support new protein synthesis, thereby linking targeted protein degradation to cellular metabolism and renewal.

However, not all proteasome-generated peptides are destined for complete degradation. A subset of these cleavage products serve specific biological functions [4]. For instance, proteasomes generate antigenic peptides that are displayed on MHC class I molecules to activate cytotoxic T cells as part of immune surveillance [5]. Some peptides act as signaling mediators, engaging receptors or intracellular pathways involved in processes such as neuronal activity [6,7], proliferation [8], differentiation [8], and programmed cell death [9,10]. Others display direct antimicrobial activity, functioning as innate immune effectors that inhibit the growth of invading pathogens [11]. In addition, the proteasome may only partially degrade specific substrates to yield active fragments that regulate intracellular pathways; a prominent example is the processing of the NF-κB1 precursor (p105) into the p50 subunit, which is required for NF-κB signaling [12]. These examples underscore the diverse and indispensable roles of the proteasome affecting various aspects of biological processes, not only through the selective removal of proteins but also through the generation of functional peptides.

To ensure specificity and prevent uncontrolled proteolysis, the proteasome’s catalytic activity is compartmentalized within the central chamber of the 20S core particle (Figure 1) [1,13]. In this configuration, only substrates with intrinsically unstructured regions can pass through the narrow axial gate of the 20S, a process that can be further facilitated by activators such as PA28αβ, PA28γ, and PA200 [14,15]. In contrast, folded proteins require ubiquitin tagging to be selected for degradation. The 19S regulatory particle recognizes polyubiquitin chains and then mediates substrate deubiquitination, unfolding, and translocation into the 20S core, where proteolysis takes place.

Structurally, the 20S core is a barrel-shaped complex composed of four stacked rings: two outer α-rings and two inner β-rings, each consisting of seven homologous subunits (Figure 1A) [16]. The outer α-rings are non-catalytic and function primarily as a gated entryway, while the inner β-rings house the proteolytic machinery. Among the seven β-subunits, only three, β1, β2, and β5, possess proteolytic activity in eukaryotes [16]. These catalytic subunits constitute the active centers of proteolysis.

The three catalytically active β-subunits of the proteasome are traditionally classified as chymotrypsin-like (β5), trypsin-like (β2), and caspase-like (β1), based on their cleavage preferences (Figure 1A) [17]. However, this categorization, rooted largely in assays using fluorogenic peptide substrates, has often obscured the versatility of the proteasome. Early biochemical studies hinted at broader cleavage capabilities, and more recent comprehensive analyses have reinforced that these classical labels represent dominant but not exclusive preferences. In this review, we examine the proteolytic sites of the proteasome in detail, with particular emphasis on the biochemical and structural features that underline their substrate specificities. We begin with a historical overview of how these proteolytic activities were first identified and characterized, and trace how cleavage specificities have diversified across organisms over the course of evolution. We then discuss the functional diversity of proteasome subtypes, their physiological relevance, the development of subunit-specific inhibitors, and computational tools for predicting cleavage sites. Finally, we highlight emerging directions aimed at refining our understanding of proteasomal cleavage landscapes.

## 2. Tracing the Discovery of Proteasome Catalytic Activities

The discovery of the proteasome in the 1980s revealed a multi-subunit “multicatalytic proteinase complex” of ~700 kDa with unusual proteolytic properties [18]. Pioneering work by Wilk and Orlowski purified this complex from bovine tissues and noted its ability to cleave a broad range of protein substrates. By 1989, it was clear that the 20S proteasome (as it came to be known) harbors multiple protease activities in one assembly [18]. Orlowski and colleagues observed at least three distinct proteolytic activities, which they operationally defined as: a chymotrypsin-like activity (preference for cleavage after hydrophobic residues), a trypsin-like activity (cleaving after basic residues such as lysine or arginine), and a peptidylglutamyl-peptide hydrolyzing (PGPH) activity (cleaving after acidic residues, particularly glutamic acid) [19]. These activities could be differentially inhibited by chemical inhibitors, hinting that they stem from separate active sites within the complex [18]. Notably, the PGPH and chymotrypsin activities were found to be stimulated by SDS or fatty acids and appeared especially important for degrading full-length protein substrates [18,20]. In 1999, the term “caspase-like activity” was introduced to replace PGPH, based on findings that the β1 subunit (PSMB6) cleaved after aspartic acid residues with significantly higher efficiency than after glutamic acid, particularly when using fluorogenic substrates [17]. Thus, the notion of the proteasome as a singular protease gave way to the understanding of a multicatalytic enzyme with multiple substrate specificities contained in one complex.

Biochemical investigations soon revealed that proteasomal specificity extended beyond the classical triad. In the early 1990s, two additional proteolytic preferences were described: cleavage after branched-chain amino acids (BrAAP) and small neutral amino acids (SNAAP), both identified using specialized fluorogenic substrates and selective inhibitors [21,22]. Subsequent mutational analyses attributed these activities primarily to the β5 subunit, suggesting that this catalytic site possesses a more versatile specificity than initially recognized [23]. Taken together, these findings supported the view that under suitable conditions, the 20S proteasome is capable of cleaving after nearly every amino acid.

A major turning point in proteasome research came with the advent of high-resolution structural analyses combined with covalent inhibitors, which revealed both the chemical nature and spatial organization of the catalytic sites. In 1995 and 1997, crystallographers led by Baumeister, Groll, Löwe and Huber reported the first crystal structures of proteasome core particles, initially from the much simpler proteasome, *Thermoplasma acidophilum* and shortly thereafter from the eukaryotic *Saccharomyces cerevisiae* 20S proteasome. The latter, composed of seven distinct α-subunits and seven distinct β-subunits, was solved at 2.4 Å resolution [24,25]. The eukaryotic structure unequivocally demonstrated that each mature β-subunit (β1, β2, β5) becomes catalytically active upon autocatalytic removal of its propeptide to expose the functional N-terminal Thr1, which forms the nucleophile in peptide bond hydrolysis [25,26]. Covalent inhibitors like lactacystin were found to specifically modify the Thr1 hydroxyl group, proving that threonine is the catalytic nucleophile and not a serine as previously speculated [27]. Collectively, these findings redefined the proteasome as a threonine protease composed of distinct catalytic subunits with specialized substrate-binding pockets.

Each of the three catalytic β-subunits β1, β2, and β5, possesses a distinct substrate-binding pocket (S1), shaped by the surrounding amino acid residues (Figure 2). The S1 pocket of β5 is large and hydrophobic, accommodating bulky nonpolar side chains like phenylalanine, tyrosine, or leucine, thus conferring its chymotrypsin-like preference. β2 has an acidic pocket that favors positively charged residues such as lysine or arginine, giving it trypsin-like preference. In contrast, β1 contains a small polar pocket that preferentially binds acidic residues, glutamic acid and aspartic acid, supporting its caspase-like activity. However, these specificities are not absolute. The binding pockets are flexible enough to accommodate non-preferred residues under certain conditions. For example, β5 can cleave after small polar residues such as serine or alanine, albeit at a reduced rate, while β2 may accept neutral residues in the absence of optimal basic ones [28]. Together, these overlapping specificities allow the three catalytic sites to cover a wide range of peptide bonds [29]. In fact, experimental data increasingly support the view that between them, β1, β2, and β5 are capable of cleaving the vast majority of peptide bonds within a substrate [29]. This broad-spectrum activity enables the proteasome to efficiently degrade diverse proteins into peptides, minimizing the risk of encountering resistant cleavage sites.

Despite this intrinsic versatility, the widespread adoption of synthetic fluorogenic peptide substrates, especially those utilizing the 7-amino-4-methylcoumarin (AMC) fluorophore, contributed to a narrower functional view of the proteasome. Substrates such as Suc-LLVY-AMC (for β5), Ac-RLR-AMC (for β2), and Ac-LLE-AMC derivatives (for β1) became the gold standard for activity assays [30]. In these assays, proteasomal cleavage of the peptide releases the AMC group, producing a fluorescent signal directly proportional to enzymatic activity. Such substrates have been indispensable tools for probing proteasome activity, providing highly sensitive, rapid, and reproducible assays that have greatly advanced our understanding of catalytic site function. Their simplicity and ease of use made them especially valuable for high-throughput studies and for dissecting the contributions of individual β-subunits.

At the same time, it is important to recognize their limitations. The short length, restricted sequence diversity, and absence of structural context mean that fluorogenic peptides are only approximations of native protein substrates. As such, they do not capture the influence of flanking residues, peptide conformation, or the kinetics of substrate unfolding and engagement. Because they are designed to probe isolated catalytic pockets, fluorogenic peptides provide a focused but inherently simplified view of proteasome activity. Consequently, less frequent yet biologically relevant cleavages, such as BrAAP and SNAAP, were often underrepresented or overlooked. This recognition has fueled calls for broader and more physiologically relevant classifications of proteasome activity, incorporating cleavage specificities that extend beyond the classical triad [21,22,23,24].

The need for unbiased assays is further underscored by comparative studies in evolutionary divergent organisms. For example, work in *Trichomonas vaginalis* has shown that fluorogenic substrates optimized for the human proteasome fail to capture species-specific differences in active-site architecture and substrate preference. These findings highlight the importance of developing organism-adapted tools for both fundamental research and therapeutic development [31].

Recent progress in this direction includes several mass spectrometry (MS)–based methods that directly identify proteasome-derived peptide products, thereby complementing fluorogenic assays and enabling condition-, tissue-, or subtype-specific conclusions about proteasome function. Multiplex Substrate Profiling by Mass Spectrometry (MSP-MS) employs diverse synthetic peptide libraries to map prime and non-prime subsite preferences (P4–P4′) in a single reaction, generating sequence motifs and specificity landscapes [32]. Mass spectrometry Analysis of Proteolytic Peptides (MAPP) identifies proteasome-cleaved peptides that are captured inside or near cellular proteasomes [33]. It relies on reversible crosslinking together with immunoprecipitation of cellular proteasomes prior to reverse-phase isolation and elution of captured peptides. Intact Degradomics Mass Spectrometry (ID-MS) enables real-time identification of generated peptides, as well as uncleaved and partially cleaved protein substrates, revealing the processive nature of protein proteasomal degradation [34]. Although these methods are more technically demanding than fluorogenic peptide assays, they provide a comprehensive view of proteasome activity, better reflecting the complexity of its cleavage specificity.

## 3. From Uniform to Specialized: Evolution of Proteasome Catalytic Sites

In *Thermoplasma acidophilum*, the 20S proteasome is composed of 14 identical β-type subunits, all catalytically active. Early characterizations have identified a singular chymotrypsin-like activity, primarily cleaving after hydrophobic residues [13]. Later studies revealed additional activity: low-level PGPH and trypsin-like cleavage, detectable under certain conditions, implying that the single active site has inherent but limited substrate flexibility [35,36,37]. Nevertheless, proteolytic preferences remain highly biased towards cleavage after hydrophobic amino acids.

In *Rhodococcus*, a representative actinomycete bacterium, subunit composition diversified, leading to increasing functional complexity. Its 20S proteasome is composed of two distinct α-type and two distinct active β-type subunits, a marked departure from the homomeric subunit composition seen in archaeal proteasomes [35,37]. Despite this increased heterogeneity, the *Rhodococcus* proteasome primarily exhibits chymotrypsin-like activity, preferentially cleaving after hydrophobic residues, with no strong evidence of additional peptidase activities under standard in vitro conditions [38]. This intermediate level of diversification offers a comparative framework for understanding more elaborate specialization observed in eukaryotic proteasomes.

In the eukaryotic 20S proteasome, gene duplication and subfunctionalization produced seven distinct β-subunits, of which only three β1, β2, and β5, retain catalytic activity [39,40]. This evolutionary refinement toward functional specialization not only expanded the range of cleavable substrates but also enabled eukaryotic proteasomes to assume more nuanced and tightly regulated roles in cellular physiology, especially within the context of complex multicellular organisms.

Phylogenetic and structural analyses reveal that β1, β2, and β5 are more closely related to each other than to the non-catalytic β-subunits, supporting the notion that they originated from a common ancestral gene [39]. Their early divergence marked a critical step in the evolution of functional diversity within the proteasome. In jawed vertebrates, this evolutionary trajectory advanced with the development of inducible paralogs, β1i (PSMB9), β2i (PSMB10), and β5i (PSMB8), which assemble to form the immunoproteasome (Figure 1B) [39,41]. These alternative subunits are not found in lower eukaryotes such as yeast, flies, or nematodes, indicating that the immunoproteasome is a vertebrate-specific innovation tailored to enhance antigen processing and immune defense [42]. Concurrently, a separate gene duplication event led to the emergence of β5t (PSMB11), a unique catalytic subunit incorporated into the thymoproteasome and expressed exclusively in cortical thymic epithelial cells (Figure 1E) [43]. Phylogenetic analyses cluster β5, β5i, and β5t within a distinct β5-derived lineage, indicating that they diverged early during evolution, where β5t appears to have evolved more rapidly than its paralogs, suggesting active evolutionary pressure in thymic contexts [42].

Together, these evolutionary developments highlight how diversification of the catalytic β-subunits enhanced the proteasome’s proteolytic capacity and enabled adaptation to specialized physiological roles. In the following section, we examine how this diversification gave rise to distinct proteasome variants, including the immunoproteasome, intermediate proteasome, thymoproteasome, and sperm-specific proteasome (spermatoproteasome) and how their association with regulatory particles further modulates catalytic output. Our perspective centers on how these specialized forms shape proteolysis across diverse biological contexts.

## 4. Proteasome Variants and Regulatory Particles: Fine-Tuning Degradation for Specialized Functions

### 4.1. Immunoproteasome

The immunoproteasome exemplifies how vertebrates have refined proteasome catalysis to support immune-specific functions. In jawed vertebrates, pro-inflammatory cytokines such as interferon-γ (IFN-γ) or oxidative stress signals, induce the replacement of the constitutive catalytic subunits β1, β2, and β5 with their inducible counterparts, β1i, β2i, and β5i, resulting in the assembly of the immunoproteasome (Figure 1B) [44].

Functionally, the incorporation of these inducible subunits adjusts the proteolytic profile of the proteasome. β1i shows reduced cleavage after acidic residues while favoring cleavage after neutral or hydrophobic residues. β2i retains a specificity profile similar to β2 but operates with distinct kinetic properties. In contrast, β5i enhances chymotryptic activity with broader specificity for hydrophobic amino acids like leucine and phenylalanine, compared to its constitutive counterpart [28,34,45]. These shifts collectively result in the preferential generation of peptides with hydrophobic or basic C-terminal residues, characteristics that enhance their compatibility with MHC class I molecules [28]. Moreover, differences in cleavage rates give rise to changes in the abundance and dominance of specific peptide species [29,34,45].

At the molecular level, sequence alignment shows that β1i contains two amino acid substitutions in comparison to the constitutive β1 subunit, which reduce the size of the S1 pocket shifting it from positively charged to more neutral. This structural change explains the significantly reduces caspase-like activity of β1i (Figure 2) [46]. X-ray crystallographic analysis further shows that while the S1 pocket of β5 favors small hydrophobic residues (e.g., alanine, valine), the corresponding pocket in β5i is significantly enlarged, allowing efficient cleavage after bulky hydrophobic residues such as tyrosine, tryptophan, and phenylalanine (Figure 2) [47]. These structural adaptations position β5i as a key contributor to the generation of MHC class I compatible epitopes during immune activation [48]. This is supported by in vivo studies showing that mice lacking β1i, β2i, and β5i present significantly fewer MHC class I bound peptides, and lose many low-abundance epitopes, resulting in impaired CD8+ T cell priming, and reduced antiviral immunity [49].

Beyond its well-characterized role in antigen processing, the immunoproteasome is increasingly recognized for broader physiological functions. It is expressed in non-hematopoietic tissues such as cortical thymic epithelial cells and pancreatic β cells, particularly under inflammatory or oxidative stress conditions [50]. Under oxidative conditions, immunoproteasome activity is often upregulated, enhancing the degradation of oxidatively modified proteins and preventing their aggregation [51]. This proteolytic adaptation protects cells from reactive oxygen species toxicity, as evidenced by β1i knockout models that accumulate oxidized, polyubiquitinated proteins and exhibit elevated ROS levels [52]. In addition, the immunoproteasome also supports cellular differentiation and proliferation. For instance, β5i knockdown in adipocytes impairs adipogenesis in vitro, and β5i loss-of-function in mice or patients results in reduced adipose tissue development [53]. Similarly, pharmacological inhibition of β5i blocks CD4^+^ T cell differentiation and cytokine synthesis and has shown therapeutic benefit in arthritis models by reducing bone erosion and joint inflammation [54]. In addition, immunoproteasome activity is essential for T cell expansion and survival during viral infection [55], while selective inhibition of β1i decreases tumor burden in a mouse colon adenocarcinoma model [56]. Collectively, these findings underscore the multifaceted roles of the immunoproteasome in health and disease, highlighting functions that extend well beyond immune surveillance [57].

### 4.2. Intermediate Proteasome

Proteasomes termed “intermediate” consist of a mix of constitutive and immuno-specific catalytic subunits, through incomplete or selectively regulated assembly [58]. These mixed proteasomes appear mostly under mild inflammatory conditions or in cells that selectively express individual immuno-subunits in an asymmetric manner [59,60]. The most common form of intermediate proteasome in vivo, is the so-called single intermediate proteasome (SIP), which contains only the β5i subunit alongside constitutive β1 and β2 subunits (Figure 1C). SIPs can account for up to 50% of the total proteasome pool depending on the tissue or cell type. Double intermediate proteasomes (DIP), which contain both β1i and β5i (with constitutive β2), have also been identified (Figure 1D) [61,62]. Intermediate forms containing only β1i and/or β2i appear under conditions where β5i is depleted, indicating that β5i displays the highest incorporation efficiency and can integrate even in the absence of β1i or β2i, enabling partial immunoproteasome configurations.

Each 20S proteasome contains two inner β-rings, meaning that every β subunit occurs in duplicate. This architecture permits the formation of asymmetric intermediate proteasomes, in which one β-ring carries constitutive catalytic subunits while the other incorporates immune-type subunits. Such hybrid 20S particles have been reported, for example, proteasomes containing both β5 and β5i were detected in IFN-β–treated MIN6 cells [63], and complexes containing both β1 and β1i were isolated from the cytoplasmic, nucleoplasmic, and microsomal fractions of HeLa cells cultured in the presence of IFN-γ [59]. These asymmetric assemblies exhibit distinct proteolytic properties within a single particle, expanding the functional diversity of the proteasome and enabling fine-tuned adaptation to specific cellular environments.

Functional studies that compared degradation by the constitutive, immuno-, and intermediate proteasomes, showed that intermediate subtypes generally display intermediate cleavage properties of their chymotrypsin-like and trypsin-like activities. In line with the enhanced chymotrypsin-like activity of the β5i, SIP, DIP and pure immunoproteasomes were shown to display enhanced chymotrypsin-like activity, compared to the constitutive proteasome. Similarly, β2i induced elevated trypsin-like activity. Incorporation of β1i into the DIP to form the full immunoproteasome significantly reduced the caspase-like activity [64]. These findings underscore the unique catalytic profiles of SIPs and DIPs, which exhibit differential efficiencies toward specific cleavage sites [62]. Interestingly though, constitutive proteasomes, immunoproteasomes, and intermediate proteasomes were shown to have a similar capacity to degrade ubiquitylated proteins [48]. This indicates that their functional distinctions are more relevant to specialized roles, such as antigen processing and immune modulation, rather than baseline proteostasis.

Intermediate proteasomes are not restricted to immune cells; they are constitutively expressed in various tissues and cell types, although their relative abundance varies across locations [61,62]. Their expression is upregulated under low-grade inflammation, such as exposure to sub-stimulatory concentrations of IL-1β. This inducibility allows cells to modulate proteasome activity in response to stress or metabolic demand, without activating a full inflammatory program [44,65].

By modulating peptide cleavage preferences, intermediate proteasomes can differentially change the pool of peptides available for MHC class I presentation, independent of the full immunoproteasome. This functional versatility may be particularly relevant in contexts such as self-tolerance, immune surveillance, and autoimmunity [46,65]. For example, in pancreatic β cells, highly active in protein synthesis and chronically exposed to inflammatory cytokines in both type 1 and type 2 diabetes, the presence of β5i-containing proteasomes suggests a role in regulating proteostasis and immune signaling [65]. The ability of intermediate proteasomes to influence the immunopeptidome in non-professional antigen-presenting cells raises important implications for disease pathogenesis in autoimmune settings.

Emerging evidence also supports a role for intermediate proteasomes in cancer biology. These mixed proteasomes are constitutively expressed in a range of solid tumors and hematologic malignancies, particularly in cells experiencing chronic inflammation or proteotoxic stress [61]. Notably, overexpression of β5i-containing intermediate proteasomes has been linked to increased resistance to proteotoxic stress and, consequently, to reduced sensitivity to chemotherapy [50]. In summary, intermediate proteasomes provide a dynamic means of fine-tuning proteasome activity between constitutive and immunoproteasome states. By enabling tissue- and context-specific regulation of proteolysis and antigen presentation, they contribute to both cellular adaptability and immune system diversity.

### 4.3. Thymoproteasome

In cortical thymic epithelial cells (cTECs), the constitutive β5 chymotrypsin-like subunit is replaced by a unique paralog, β5t (PSMB11), while β1i and β2i remain from the immunoproteasome repertoire [66]. Functionally, β5t confers reduced chymotrypsin-like activity relative to β5 or β5i (Figure 1E) [39]. This attenuation reflects substitutions that make the β5t S1 pocket more hydrophilic than in β5/β5i, weakening recognition of bulky hydrophobic P1 side chains (P1 is the residue immediately N-terminal to the scissile bond) and thus reducing the production of peptides with classical hydrophobic C-termini (Figure 2) [67]. At the same time, trypsin-, and caspase-like activities remain largely unchanged, as β1i and β2i retain their S1 pocket architecture and functionality.

Consequently, the thymoproteasome yields a distinctive shift towards hydrophilic or basic C-terminal residues, reducing the abundance of strongly hydrophobic peptides [68]. This adaptation results in altered peptide generation, which is distinct from that generated by either β5 or β5i. Interestingly, its lower MHC class I binding affinity supports the positive selection of CD8^+^ thymocytes, by promoting T-cell receptor interactions of intermediate strength, which are too weak for survival signals but not too strong to trigger clonal deletion, and result in generation and maturation of a diverse pool of self-tolerant CD8^+^ T-cells. In mice, deficiency in β5t displays reduced numbers of mature CD8^+^ T-cells, accompanied by an altered T-cell receptor repertoire, which displays higher-affinity interactions and reduced clonal diversity [69].

Assembly of the thymoproteasome is stringently regulated in cTECs. Unlike β5i, which can incorporate independently in intermediate immunoproteasomes, β5t is only incorporated after β1i and β2i have been assembled [70]. This dependency prevents irregular β5t inclusion in non-thymic tissues and ensures that the thymoproteasome formation is restricted to a tissue environment where they can produce the specialized peptide repertoire required for thymic selection [58]. Together, this demonstrates how evolutionary specialization of a single active subunit can reshape peptide generation, fine-tuning the output to meet the precise immunological needs of thymic T cell development.

### 4.4. Spermatoproteasome

The spermatoproteasome (s20S) is a specialized variant of the 20S proteasome found exclusively in male germ cells. It is defined by the substitution of the canonical α4 subunit (PSMA7) with the testis-specific paralog α4s (PSMA8) (Figure 1F). Although this modification does not alter the composition of the catalytic β-subunits, it has significant structural and regulatory implications that distinguish the s20S from the constitutive proteasome [71,72].

During spermatogenesis, particularly as germ cells enter later stages of differentiation, the proteasome pool progressively shifts from predominantly constitutive 20S complexes to s20S assemblies containing α4s. By the spermatid stage, s20S can account for over 80–90% of the total 20S population [73]. α4s is evolutionarily conserved across vertebrates and is essential for proper male fertility: knockout of the *PSMA8* gene in mice leads to arrest at meiotic prophase I and complete male infertility, highlighting the necessity of s20S in maintaining proteostasis during germ cell development [74,75].

Comparisons of enzymatic activities indicate that s20S displays elevated chymotrypsin-like and trypsin-like activities relative to the constitutive proteasome [72]. Beyond intrinsic catalytic differences, the α4s subunit promotes preferential assembly with specific proteasome regulators, including PA200, 19S (see Section 5), and the adaptor protein PI31, further modifying the functional landscape of the s20S complex [71,72,76]. The association of s20S with PA200 promotes acetylation-dependent proteolysis of core histones in an ATP- and ubiquitin-independent manner, during the critical chromatin remodeling step of spermiogenesis [77]. Meanwhile, association with the 19S regulatory particle aligns with the high demand for ubiquitin-dependent proteolysis during earlier meiotic stages. Hydrogen–deuterium exchange mass spectrometry and molecular dynamics simulations indicate that α4s alters the dynamics of the α-ring and enhances regulator affinity, which consequently contributes to modulation of substrate specificity by influencing substrate access, processing efficiency, and regulatory scope [72]. Thus, s20S exemplifies how subunit substitution without changing the catalytic core can reprogram proteasome function to meet specialized cellular needs [78].

## 5. Diversification and Loss in Higher Eukaryotes: Adaptive Evolution of Catalytic Subunits

The evolutionary trajectory of proteasome catalytic subunits has diverged significantly across the animal and plant kingdoms, reflecting species-specific pressures on immune function, development, and stress adaptation. A striking example of subunit loss is observed in birds: while β5t is present in most mammals, reptiles, and fish, it is absent in avians, making birds a natural knockout model lacking both immunoproteasomes and thymoproteasomes [39,43]. This coordinated loss suggests a significant evolutionary shift in thymic selection mechanisms in the avian lineage.

Conversely, in various vertebrates, catalytic diversification has expanded, particularly through duplication of immune-associated subunits like β5i and β5t. In teleosts, the largest group of bony fish, two highly divergent β5t alleles have evolved. These isoforms differ substantially in their amino acid sequences, especially in the architecture of their S1 substrate-binding pockets [43]. The β5tb subunit preserves the hydrophilic S1 configuration similar to that of mammalian β5t, which reduces the cleavage efficiency after hydrophobic residues, while the β5ta subunit has a more hydrophobic pocket, implying for a higher capacity to facilitate chymotrypsin-like activity. These differences likely reflect divergent cleavage specificities that could influence the quality of peptides produced during thymic selection.

Other diversifications in catalytic proteasome subunits were found in *Xenopus* (clawed frogs) [79], *Oryzias* (a genus of ricefishes) [80], as well as in the basal actinopterygian *Polypterus senegalus* (a ray-finned fish) [81], and sharks [82]. In these organisms, two dichotomous alleles of β5i are found, which differ at the key residue 31 within the S1 substrate-binding pocket of the mature protein, one containing phenylalanine (Phe31) and the other containing alanine (Ala31). These amino acid variants form two major functional classes: Phe31-type alleles promote chymotrypsin-like activity, and favor cleavage after bulky hydrophobic residues, which are optimal for generating MHC class I compatible peptides. Ala31-type alleles shift the preference toward elastase-like activity and cleave after smaller neutral residues (which are potentially less favorable for MHC class I binding).

Beyond vertebrates, testis-specific proteasome diversification has occurred in insects. For instance, in *Drosophila*, multiple subunits of both the 20S and 19S proteasome, particularly catalytic β2 and β5 subunits, have duplicated and are expressed exclusively in the testis [83]. These testis-specific isoforms show sequence divergence from their somatic counterparts and are thought to have roles in gametogenesis and genome surveillance, likely contributing to the protection and remodeling of the male germline.

In contrast, plants represent a distantly related evolutionary lineage in which proteasome diversification has followed a different trajectory. Phylogenomic analyses in major plant families (*Brassicaceae*, *Poaceae*, *Solanaceae*) have revealed widespread duplication events affecting both 20S catalytic subunits and 19S regulatory components [84,85]. Such duplications arising through whole-genome, tandem, or segmental events are a common feature of plant genomes and likely reflect a general strategy for enhancing adaptability in sessile organisms. Although these patterns align with the general trend of adaptation to fluctuating and complex environmental conditions, the specific functional roles of many of these plant-specific isoforms remain largely uncharacterized, offering intriguing avenues for future research.

## 6. Association with Regulatory Particles

The 20S proteasome core particle carries out proteolysis through its three intrinsic catalytic sites, yet its substrate selection, peptide product size and cleavage site preferences are heavily influenced by the regulatory particles that can cap the core. These regulators include the ATP-dependent 19S, as well as ATP-independent regulators PA200, PA28αβ, and PA28γ (Figure 1G). These regulators can bind to one or both α-rings of the 20S core and, in some cases, form hybrid proteasomes with different regulators capping each end. Each regulator confers distinct functional properties by modulating the proteasome’s proteolytic capacity in unique ways including gate opening, allosteric modulation, and changes in substrate dwell time inside the catalytic chamber. Through these mechanisms, regulatory particles fine-tune the proteasome’s catalytic output, with consequences for immunopeptidome composition, signaling, and metabolism.

### 6.1. 19S Regulatory Particle

The 19S regulatory particle (also called PA700) is the canonical regulator of the 20S proteasome. When a single 19S complex binds to one end of the 20S core, the 26S proteasome is formed. If both ends of the 20S are capped by 19S complexes, the resulting assembly is referred to as the 30S proteasome (Figure 1G). The 19S recognizes ubiquitinated substrates, removes ubiquitin chains, unfolds the substrate using its AAA+ ATPase base subunits, and translocates the substrate into the 20S core. Although it does not directly alter the cleavage specificity of the β subunits, the 19S confers substrate selectivity through its ubiquitin receptors (Rpn1, Rpn10, Rpn13) and ATP-driven translocation. Proteasomes capped by the 19S tend to generate shorter peptide fragments compared to those produced by the 20S proteasome alone [86].

### 6.2. PA28αβ

The PA28αβ complex (also known as 11S regulator), is a ring-shaped heptameric heterocomplex and the second most abundant proteasome regulator after the 19S [61]. It is constitutively expressed in lymphoid cells and immunological organs, but also in a variety of non-immune tissues such as erythrocytes, muscle, and brain. Its expression is highly induced in response pro-inflammatory cytokines. PA28αβ activates the proteasome in an ATP- and ubiquitin-independent manner by inducing gate opening through interaction with the 20S α-ring, thus enhancing peptide entry [87]. Early kinetics studies show that it can increase the maximal turnover rate (V_max_) and lower the Michaelis constant (K_m_) for peptide substrates by up to 50-fold [88]. Although it does not substantially alter the intrinsic cleavage specificities of the β1, β2, and β5 subunits, its ability to accelerate substrate processing tends to favor the production of shorter peptide fragments compared to those generated by the 26S proteasome [89].

PA28αβ binds to both the constitutive proteasome and the immunoproteasome (i20S) with comparable affinity [90]. However, because it is co-induced with the immunosubunits during inflammatory responses, its association with the immunoproteasome is favored under cytokine-stimulated conditions [91], reinforcing its traditional association with immune-modulatory proteasome functions. Despite this, the role of PA28αβ in shaping the MHC class I immunopeptidome remains debated. Some studies have shown that it does not significantly impact the generation of classical MHC class I peptides [92], due to its increased yield of shorter and more hydrophilic peptides [93], which may fall outside the preferred parameters for MHC class I loading. Interestingly, recent evidence suggests a potential role for PA28αβ in shaping the CD8^+^ T cell repertoire in the thymus, likely through its interaction with the thymoproteasome [92].

### 6.3. PA28γ

PA28γ is a homoheptameric, ring-shaped proteasome regulator predominantly localized in the nucleus and broadly expressed across all tissues and organs [94]. It binds to the α-ring of the 20S proteasome and allosterically enhances trypsin-like activity, influencing peptide cleavage without requiring ATP or ubiquitin [95]. This selectivity indicates an allosteric communication: PA28γ’s interaction with the 20S core induces conformational or dynamic changes that specifically favor the β2 active-site configuration or kinetics. Notably, these effects persist even when the proteasome gate is held open experimentally, implying that PA28γ’s stimulation of β2 is not due to enhanced substrate entry alone, but rather due to direct allosteric tuning of the active site’s catalytic parameters [96].

Phylogenetic analyses reveal that PA28γ is conserved in both jawless and jawed vertebrates, whereas PA28αβ is found only in jawed vertebrates with the notable exception of birds. This evolutionary finding suggests that *PA28γ* represents a slow-evolving gene, which laid the foundation to the more recent, faster evolving PA28αβ [97]. As a broadly and ubiquitously expressed 20S regulator, PA28γ is involved in the regulation of essential cellular processes such as cell growth, proliferation and apoptosis through chromatin organization, and responses to DNA damage [93].

### 6.4. PA200

Another ATP-independent regulator, PA200, is a large, monomeric activator localized primarily in the nucleus [98]. Cryo-electron microscopy studies have shown that the dome-like architecture of PA200 induces allosteric rearrangements of the catalytically active β-subunits. These structural shifts lead to β2 active site widening, while the β1 and β5 sites narrowed. Resulting in increased trypsin-like (β2) and decreased chymotrypsin-like (β5) and caspase-like activities (β1) in vitro [98,99]. Thus, PA200’s effects are not simply identical to other activators like PA28 or the 19S; rather it confers a unique tuning of the proteolytic active sites via its distinct structural interactions.

### 6.5. PI31

Unlike the classical activators, PI31 is a versatile proteasome regulator that operates at the levels of assembly, localization, and proteasomal dynamics [88,100]. Initially characterized as an inhibitor of 20S proteasome, more recent findings have revealed a context-dependent role. Under IFN-γ stimulation, PI31 was shown to facilitate the incorporation of the inducible β5i subunit into assembling 20S proteasomes. In the absence of PI31, immunoproteasome maturation was impaired and chymotrypsin-like activity was significantly reduced [100]. These results suggest that PI31 plays a chaperone-like role for the selective assembly of 20S immunoproteasomes.

Collectively, the selective interaction of specific proteasome subtypes with distinct regulators enables fine-tuned, tissue- and context-specific catalytic output adaptations, adding a coordinated layer of control to proteasome function and substrate processing.

## 7. Tuning Proteasome Activity Through Post-Translational Modifications

Post-translational modifications (PTMs) of 20S proteasome subunits serve as dynamic regulatory mechanisms that adjust catalytic activity in response to cellular signals and stress. Phosphorylation often enhances the catalytic activity; for example, protein kinase A-mediated phosphorylation of the 20S core (at multiple serine/threonine sites on β2, β3 and β7, as well as certain α-subunits) elevates all three peptidase activities (caspase-, trypsin-, and chymotrypsin-like) [101]. In contrast, dephosphorylation by protein phosphatase 2A reverses these effects, reducing proteasome activity [101]. Additional studies have shown that phosphorylation of both 19S and 20S subunits can either stimulate or suppress proteasome function, depending on the site being modified [101]. Acetylation also plays a role in modulating activity; for instance, acetylation of specific lysine residues in the α6 subunit (PSMA1) of the mouse myocardium was associated with increased trypsin-like activity [102].

In addition, both the 20S core and 19S regulatory particles are subject to various redox modifications, including glycoxidation, lipid peroxidation by 4-hydroxynonenal (4-HNE), carbonylation, and S-glutathionylation, most of which impair proteolytic activity [101]. For example, carbonylation of RPT3 (*PSMC4*) was shown to reduce its ATPase activity, followed by decline in ubiquitin and ATP-dependent proteolysis by the 26S proteasome [103]. Similarly, carbonylation or 4-HNE modification of the 20S proteasome were also shown to inhibit its catalytic activities [104]. In contrast, S-glutathionylation of cysteines on the α5 (PSMA5) subunit of 20S proteasomes was shown to open the gate of the 20S proteasome, enhancing its proteolytic activity [105]. Together, these modifications enable the cell to fine-tune proteasome activity by altering its efficiency and capacity without altering the catalytic subunits composition.

Proteasome function is fine-tuned not only within cells but also in extracellular environments [106]. For example, circulating 20S proteasomes in blood plasma have been reported to carry PTMs such as cysteinylation and glutathionylation on both α and β subunits, with particularly high frequency on catalytic subunits of both constitutive and immunoproteasomes [107]. These modifications are linked with the enhanced caspase-like activity of the circulating proteasome relative to its intracellular counterpart, suggesting that PTMs can modulate the activity of extracellular complexes. Together, these findings highlight the remarkable versatility of the proteasome and the multiple regulatory layers that govern its function both inside and outside the cell.

## 8. Proteasome Dynamics Across Physiological Conditions

In the sections above, we detailed the structural diversity of proteasome subtypes as well as their functional modulation through association with various regulatory particles. Together, these variants and regulators provide the proteasome system with remarkable flexibility to meet tissue- and context-specific demands. In the following section, we turn our attention to how this functional versatility translates into changes in proteolytic activity under distinct physiological conditions. Specifically, we explore how differentiation, inflammatory signals, oxidative stress, and aging influence proteasome composition, catalytic output, and substrate specificity.

### 8.1. Differentiation

During cellular differentiation, the proteasome system is dynamically remodeled to meet the functional demands of specialized cell types. Both proteasome activity and composition undergo significant changes as progenitor cells become specialized. For example, human embryonic and pluripotent stem cells exhibit elevated proteasome activity compared to their differentiated counterparts, a feature essential for maintaining pluripotency and self-renewal [108]. When proteasome activity is reduced, these cells lose their ability to differentiate into neural lineages and instead express elevated levels of markers associated with endodermal, mesodermal, and fibroblastic fates [108]. Similarly, in the context of fibrotic lung tissue, the activation and differentiation of pro-fibrotic myofibroblasts requires increased assembly and activity of 26S proteasomes, highlighting the proteasome’s involvement in tissue remodeling and stress-responsive cell fate transitions [109].

Changes in proteasome subunit composition also contribute to differentiation-associated remodeling. Immunoproteasome subunits have been implicated in multiple differentiation pathways. In C2C12 skeletal muscle cells, differentiation is accompanied by elevated mRNA and protein levels of both constitutive and immunoproteasome subunits, as well as increased proteolytic activity of the 20S and 26S complexes [110]. Functional studies show that knockdown of β1i or treatment with immunoproteasome-specific inhibitors reduces 26S activity and impairs myogenic differentiation [110]. Another example comes from differentiating mouse embryonic stem cells which exhibit upregulation of the immunoproteasome subunit β5i and the PA28αβ activator, even in the absence of immunocytokine signaling [111].

Differentiating cells can also alter proteasome composition by engaging alternative activators. Transient silencing and overexpression revealed that PA200 acts as a negative regulator of myofibroblast differentiation of human but not mouse cells [112]. A recent study has shown that PA28γ-knockout mice had low bone mass coupled with excessive marrow fat, indicating a shift toward adipocyte formation at the expense of osteoblasts. Also, knockout of PA28γ in bone marrow stromal cells suppressed osteogenic differentiation and enhanced adipogenic differentiation, whereas overexpressing PA28γ had the opposite effect [113].

### 8.2. Inflammation

Inflammatory conditions provoke a rapid adaptive change in proteasome composition, most prominently via the induction of immunoproteasomes. Exposure to proinflammatory cytokines like IFN-γ causes cells to replace the constitutive proteasome β-subunits with inducible immunoproteasome subunits (β1i, β2i, β5i), which alters the proteolytic specificity of the proteasome [114,115,116,117,118]. In parallel, IFN-γ triggers upregulation of the proteasome activator PA28αβ, further modulating proteasome activity by opening the 20S core to favor the production of shorter peptides suited for antigen presentation [89,118,119]. Notably, this cytokine-driven proteasome remodeling is tightly regulated and transient: immunoproteasomes have a shorter half-life, and once the inflammatory signals subside, cells revert to primarily assembling constitutive proteasomes [120].

### 8.3. Oxidative Stress

Oxidative stress leads to the accumulation of oxidized and misfolded proteins, placing a significant burden on the cellular protein quality control systems. One of the early responses to oxidative stress is the disassembly of the 26S proteasome into its 20S core and 19S regulatory particles [121,122]. This occurs because the 26S complex is more sensitive to oxidative damage, whereas the 20S core particle is more stable and retains activity under these conditions [123].

In addition to post-translational effects, oxidative stress also induces transcriptional upregulation of 20S core subunits, while expression of the 19S regulatory components remains largely unchanged [124]. Concurrently, oxidative stress impairs the ubiquitylation and deubiquitylation machinery as several E3 ligases and deubiquitinases become inactivated [125,126], and partially unfolded, oxidized proteins are often not preferentially ubiquitinated [127]. This disruption, along with ATP depletion during oxidative stress, limits the functionality of the ubiquitin–proteasome system. In contrast, the 20S proteasome operates independently of ATP and ubiquitin, making it well-suited to degrade structurally compromised proteins, under stress conditions [128]. These findings support the notion that 20S-mediated proteolysis becomes the dominant mechanism for clearing damaged proteins during oxidative stress [96].

Moreover, several studies have demonstrated that oxidative stress also triggers the induction of alternative proteasome regulators, including PA28αβ, PA28γ, and PA200 as well as the immunoproteasome [124,129,130], even in the absence of IFN-γ signaling. These alternative forms are thought to further enhance the degradation of oxidized proteins, supporting an adaptive response aimed at restoring proteostasis under oxidative conditions [131].

### 8.4. Aging

Advancing age is consistently associated with a decline in proteasomal proteolytic capacity across various tissues. Over the past several years, numerous studies have reported reduced catalytic activity of the proteasome in aged cells from the brain [132,133,134], liver [134,135], heart [136], muscles [137], epidermal cells [138], and from T-lymphocytes [139]. Mechanistically, this age-related decline is attributed to several factors: reduced expression of proteasome subunits, altered subunit composition or assembly efficiency, and partial disassembly or inactivation of proteasome complexes, often driven by interactions with age-associated protein aggregates.

One hallmark of this decline is the preferential disassembly of the 26S proteasome, resulting in an increased ratio of free 20S cores in aged tissues across multiple species [132,140,141,142]. While the 20S core retains some ability to degrade damaged proteins, the loss of 26S compromises the degradation of polyubiquitinated substrates, leading to the accumulation of ubiquitinated and oxidized proteins. This buildup not only reflects impaired proteolysis but can also further hinder proteasome activity, promoting a vicious cycle of proteotoxic stress contributing to age-related cellular dysfunction [143].

## 9. Computational Prediction of Proteasomal Cleavage Sites

Computational algorithms have been developed to predict proteasomal cleavage sites and the peptides they generate. These approaches consistently support a central experimental finding: although the β1, β2, and β5 subunits each exhibit preferred cleavage chemistries defined by their S1 pockets, the proteasome retains the capacity to cut after nearly every amino acid. Early efforts to formalize this observation into predictive tools date back to Holzhütter et al. (1999), who sought to move beyond models that relied solely on the identity of the P1 residue [144]. Using peptide-cleavage data available at the time, they introduced a statistical framework to identify cleavage-determining motifs, noting that the actual P1 residues often diverged from those associated with the classical chymotrypsin-, trypsin-, or caspase-like activities. This first static model estimated cleavage likelihoods based on side-chain properties rather than individual residues. The approach was later expanded into a kinetic model capable of predicting double-cleavage fragments and thus the actual peptides produced [145]. Consistent with biochemical findings, leucine was most frequently observed at cleavage sites, while the highest catalytic rates were predicted for residues such as cysteine, glycine, glutamic acid, and tryptophan.

Building on this foundation, the concept was then extended to better capture the combined action of the three catalytic sites of the yeast 20S proteasome [146]. By training on a larger dataset, a one-layer network-based model was developed that incorporated broader sequence context, revealing preferences such as phenylalanine, leucine, and tryptophan in P1. This framework was later made publicly available as PAProC, which integrated proteasomal cleavage prediction with MHC class I binding analysis [147].

The introduction of artificial neural networks in 2002 marked a further step forward. Hence models were trained on either in vitro cleavage data or naturally processed MHC class I ligands, resulting in NetChop, with distinct models trained for constitutive versus immunoproteasome-like specificities [148]. A later update, NetChop 3.0, incorporated ensembles and richer sequence encodings, improving sensitivity by about 10% and reducing false positives by ~15% [149]. Both versions emphasized that proteasomal cleavage does not follow simple “P1 residue rules”: constitutive proteasomes show higher cleavage after hydrophobic residues (leucine, phenylalanine, tyrosine) and maintain cleave after acidic residues (aspartic acid, glutamic acid), whereas immunoproteasomes favor hydrophobic residues more strongly and reduce cleavage after acidic residues. Benchmarking studies subsequently established NetChop as a standard reference for evaluating new prediction tools [150].

In parallel, other tools were developed to complement and expand these approaches. ProPred1 [151], for instance modeled cleavage preferences of both constitutive and immunoproteasomes alongside MHC class I binding, using weight matrices derived from 12-mer peptide cleavage data [152]. Its cleavage module was later incorporated into nHLAPred [153], a hybrid framework that combined matrix-based and neural network methods for MHC class I binding while retaining the same proteasome-cleavage filter. Similarly, PCleavage [154] applied a support-vector-machine approach to predict constitutive and immunoproteasome sites, offering another branch within this growing family of predictors.

More recently, NetChop has been directly compared to the novel PCPS/iPCPS platform, which employs n-gram models to predict constitutive and immunoproteasome cleavage [155]. In head-to-head comparisons, PCPS achieved higher sensitivity but slightly lower specificity than NetChop. The updated iPCPS further expanded functionality by directly outputting candidate CD8^+^ T-cell epitopes and allowing users to model combined proteasome types or exclude peptides with predicted internal cleavage sites (cleavage site not in the C-terminus of the epitope), underscoring the field’s ongoing efforts to refine both accuracy and biological relevance.

Building on this momentum, Pepsickle emerged as a next-generation open-source platform that integrates gradient-boosted classifiers with deep neural networks [156]. Trained on the largest dataset to date, including 35 in vitro digestion studies alongside extensive in vivo ligand collections, Pepsickle provides dedicated models for in vitro constitutive and immunoproteasomes, as well as a high-performing in vivo (epitope-based) model. Feature analysis identified trends such as preferences for smaller or less-hydrophobic residues at P1, low conformational entropy at P1, and increased polarity at P2. On benchmark tests, Pepsickle consistently outperformed earlier tools including NetChop, PCPS, and PCleavage, highlighting the advances possible with larger training sets and modern machine-learning methods.

Since all existing prediction tools are trained on experimentally derived cleavage data, they inherently capture the broader sequence context influencing proteasomal activity. This has challenged the earlier assumption that cleavage is determined exclusively by the P1 residue and instead underscores the roles of flanking residues, sequence motifs, and proteasome subtype. As a result, these models not only reinforce experimental evidence for broad substrate flexibility but also offer a valuable framework for understanding how sequence context modulates proteasomal cleavage outcomes.

## 10. Targeting the Proteasome Catalytic Sites: Chronological Advances and Clinical Applications

The identification of selective proteasome inhibitors in the early 1990s provided critical molecular tools for dissecting the functional role of the proteasome system. Among the earliest compounds, lactacystin, a non-peptidic natural product isolated from *Streptomyces*, was reported in 1991 as the first small-molecule inhibitor of the proteasome and rapidly became a foundational reagent in biochemical studies of proteolysis [157]. This was soon followed by peptide-based aldehyde inhibitors such as MG115 and MG132, as well as epoxomicin, a natural product inhibitor that demonstrated high potency and specificity for the proteasome’s catalytic sites [158]. These molecules facilitated the functional mapping of proteasomal subunits and allowed researchers to link proteasome activity with cellular processes.

Notably, the majority of these early inhibitors, and nearly all clinically relevant derivatives developed subsequently, were designed to target the β5 subunit. The β5 active site is generally considered rate-limiting for bulk protein degradation, such that its inhibition disproportionately affects overall proteolytic capacity. This property rendered β5 a primary pharmacological target, and consequently, both first-generation tool compounds and approved therapeutic agents were engineered to preferentially block this site [158].

The transition from research tool to clinical drug was exemplified by the development of bortezomib, a boronate-based reversible β5 inhibitor approved in 2003 for the treatment of multiple myeloma and mantle cell lymphoma [159]. Bortezomib demonstrated that selective inhibition of the proteasome could effectively induce proteotoxic stress and apoptosis in tumor cells, revolutionizing treatment strategies for hematologic malignancies. Subsequently, second-generation inhibitors such as carfilzomib (a covalent epoxyketone) and ixazomib (an oral prodrug) were developed, offering improved pharmacokinetics and specificity, while continuing to exploit the β5 subunit as a therapeutic target [159].

While β5 inhibition remains central to clinical strategies, subsequent research has explored the roles of other catalytic sites [3]. Inhibitors targeting β1 and β2 subunits have been developed as complementary approaches. Although inhibition of β1 or β2 alone is generally insufficient to induce cytotoxicity, their combined inhibition with β5 can synergistically enhance proteasome blockade [160,161]. For instance, β1-selective inhibitors have been shown to sensitize cancer cells to β5-targeted drugs [161], and the β2-specific inhibitor LU-102 significantly augmented the cytotoxic effects of bortezomib and carfilzomib in multiple myeloma models [160]. These findings support the rationale for dual- or triple-site inhibition to overcome drug resistance or enhance antitumor efficacy.

Parallel research has focused on the immunoproteasome, which is frequently upregulated in cancers and inflammatory conditions, driving the development of subunit-selective inhibitors directed at its unique catalytic components, β1i, β2i, and β5i [162]. The β5i-selective inhibitor ONX-0914 (also known as PR-957) was one of the first compounds to demonstrate anti-inflammatory efficacy by dampening cytokine production and autoimmune responses in preclinical models [54]. Similarly, UK-101, a selective β1i inhibitor, induced apoptosis in prostate cancer cells and reduced tumor burden in vivo [163]. These agents offer the potential for targeted immunomodulation with reduced systemic toxicity. Several immunoproteasome inhibitors, such as KZR-616 (dual β1i/β5i inhibitor), are now in clinical trials for autoimmune disorders, further underscoring the therapeutic potential of immunoproteasome-selective targeting [164].

More recently, a novel direction has emerged in the form of species-specific proteasome inhibitors, aimed at targeting pathogens with structurally unique proteasomes. *Mycobacterium tuberculosis* was a pioneering example: *M. tuberculosis* is one of the few bacteria that possess a proteasome, and its 20S core particle has distinct subunits (only one type of β subunit, versus seven types in mammals) [165]. In 2009, researchers identified oxathiazolone compounds (e.g., GL5 and HT1171) that act as suicide-substrate inhibitors highly specific for the mycobacterial proteasome [165]. Remarkably, these inhibitors were >1000-fold more potent against the *M. tuberculosis* proteasome than the human proteasome, and they killed non-replicating *tuberculosis* bacteria in vitro while showing no toxicity to mammalian cells at effective doses [165].

Building on this concept, the proteasomes of parasitic protozoa have also been successfully targeted. The compound GNF6702 selectively inhibits the proteasome in kinetoplastid parasites (*Trypanosoma, Leishmania*) by exploiting subtle differences in the β5 active site [166]. In murine models of leishmaniasis, Chagas disease, and sleeping sickness, GNF6702 achieved complete parasitic clearance without affecting mammalian proteasomes. An optimized analog, LXE408, has since progressed to human clinical trials [167]. Similarly, the *Plasmodium falciparum* proteasome, responsible for the most severe form of human malaria, has been specifically targeted. Inhibitors such as TDI-8304, a macrocyclic peptide, have been shown to selectively bind the parasitic 20S proteasome complex over its human counterpart [168]. These findings underscore the feasibility of developing pathogen-selective proteasome inhibitors by leveraging unique structural features within the catalytic cores of microbial proteasomes.

## 11. Summary and Future Perspectives

Our understanding of proteasome biology has expanded considerably, revealing it as a modular and highly adaptable proteolytic system. Its activity is shaped by the composition of its catalytic β-subunits, the spectrum of associated regulatory particles, PTMs, and evolutionary diversification across species and tissues. Despite these advances, much of the current knowledge on proteasome specificity originates from assays with short fluorogenic peptides. While these reagents have been critical for mapping site-specific activities, they provide only a narrow view of proteasome function and may have obscured its full substrate breadth. Evidence now, as early studies also demonstrated, has shown that the β1, β2, and β5 subunits, though classically categorized by caspase-like, trypsin-like, and chymotrypsin-like preferences, possess overlapping and more extensive cleavage abilities than these labels suggest.

Progress will require moving beyond simplified model substrates and instead applying biologically relevant proteins and peptides that preserve native sequence context, structural constraints, and physiological modifications. Such approaches are critical for defining how proteasomes in different organs, developmental stages, or pathological conditions generate unique repertoires of cleavage products. Within these repertoires, particular attention should be given to peptides that are not rapidly degraded to amino acids but persist to perform regulatory or signaling roles. These functional fragments have been implicated in immune surveillance, intracellular signaling networks, stress adaptation, neuronal communication, and antimicrobial defense [5,6,7,8,9,10,11], highlighting their potential biological and clinical significance.

High-resolution structural analysis will be equally important for the next phase of discovery. Comparative mapping of catalytic pocket architectures across β-subunits, proteasome subtypes, and species can elucidate how fine structural distinctions translate into differences in substrate selection and cleavage patterns. This level of detail will be invaluable for the rational design of targeted inhibitors. Work on the *Mycobacterium tuberculosis*, *Leishmania* and *Plasmodium falciparum* proteasome has already demonstrated that species-specific active site geometries can be exploited for selective drug development [165,166,167,168]. Extending similar structure-guided strategies to other pathogens, tissue-restricted proteasomes, and disease-associated variants could yield a new class of precision therapeutics, tailored to specific proteasomal forms while sparing others.

Integrating degradomics of native substrates, functional peptidomics, structural biology, and medicinal chemistry will be important for building a more complete picture of proteasome function. This combined approach can deepen our understanding of its roles in health and disease and help guide the development of targeted therapies.

## Figures and Tables

**Figure 1 biomolecules-15-01524-f001:**
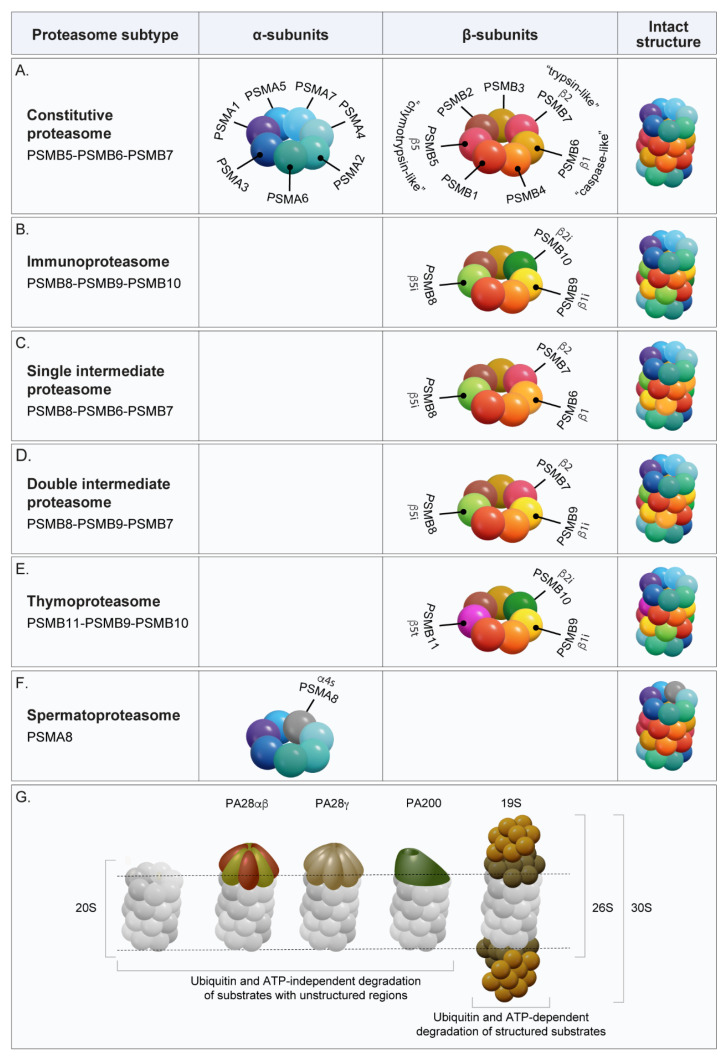
Function and architecture of proteasome subtypes. (**A**) The constitutive 20S proteasome is composed of four stacked heptameric rings: two outer α-rings (colored in cool colors) and two inner β-rings (colored in warm colors). Among the β-subunits, PSMB5 (β5), PSMB6 (β1), and PSMB7 (β2) are catalytically active. (**B**) In the immunoproteasome, the three constitutive catalytic subunits are replaced by the alternative and inducible catalytic subunits: PSMB8 (β5i), PSMB9 (β1i) and PSMB10 (β2i). Intermediate proteasomes contain a mixture of constitutive and immuno-subunits, with either one (PSMB8; (**C**)) or two (PSMB8 and PSMB9; (**D**)) of the three inducible catalytic subunits. (**E**) The thymoproteasome incorporates the unique catalytic subunit PSMB11 (β5t), together with the immunoproteasome subunits PSMB9 and PSMB10. (**F**) The spermatoproteasome is distinguished by the testis-specific α-subunit PSMA8 (α4s). Panels (**A**–**F**) are arranged as a table summarizing the α- and β-ring compositions of different proteasome subtypes; blank cells indicate no change in subunit composition relative to the constitutive 20S proteasome. (**G**) The 20S proteasome can associate with regulatory activators on one or both ends of its barrel-shaped structure. PA28α/β (olive/copper) and PA28γ (beige) form hetero- or homo-heptameric rings, while PA200 (green) is monomeric. Ubiquitin- and ATP-dependent degradation is mediated by the 26S and 30S proteasomes, formed when one or two 19S regulatory particles bind to the 20S, respectively. The 19S is composed of a lid (gold) and a base (brown). Hybrid proteasomes can also form when the 20S is capped with different activators, typically a 19S particle together with PA28α/β or PA28γ.

**Figure 2 biomolecules-15-01524-f002:**
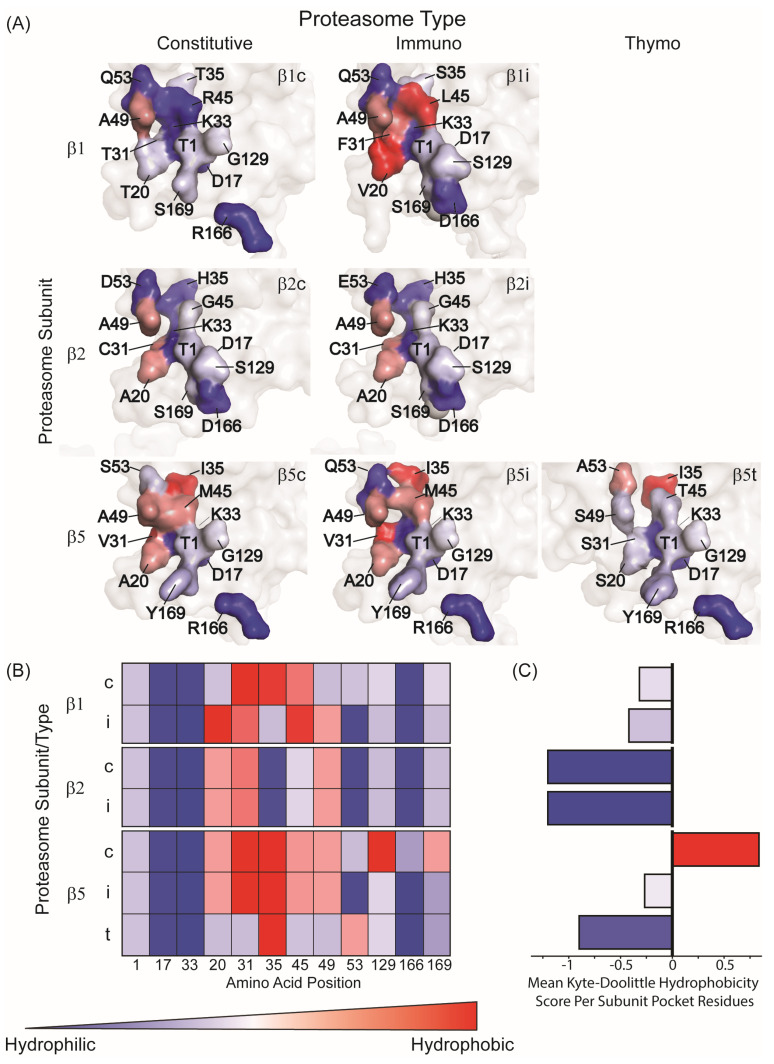
Hydrophobicity landscape of the S1 catalytic pocket across proteasome subtypes. Color scheme (all panels): hydrophilic residues, blue; neutral, gray; hydrophobic, red. (**A**) Surface views of β1, β2, and β5 from constitutive, immuno, and thymoproteasomes (based on PDB 6RGQ and 6E5B). The thymoproteasome β5 (β5t) was modeled with AlphaFold and fitted to the 6E5B immunoproteasome template. Residues comprising the S1 pocket are colored and labeled; the remaining surface is shown in semi-transparent gray. Per-residue Kyte–Doolittle hydrophobicity values were normalized to 0–100%. (**B**) Heat map of Kyte–Doolittle hydrophobicity values for S1-pocket amino acid positions (columns) across subunits (rows); c, i, and t denote constitutive, immuno-, and thymoproteasome subunits, respectively. (**C**) Mean S1-pocket Kyte–Doolittle indicating hydrophobicity score for each subunit, averaged over all pocket residues.

## Data Availability

No new data were created or analyzed in this study.

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
