# Peer review of "Cleaving Expectations: A Review of Proteasome Functional and Catalytic Diversity"

_biomolecules, 2025, doi:10.3390/biom15111524_

Round 1

Reviewer 1 Report

Comments and Suggestions for Authors

Summary: This is an excellent review article on proteasome function and diversity. I enjoyed reading it.  I plan to provide the final published version to any new students in my lab so they get a complete overview of the differences between each of the human proteasome isoforms.

I have a few minor comments that should be addressed:

Figure 1, it would be helpful to label the left boxes as “α-subunits” and the right boxs as “β-subunits”’

Figure 1F, place the α-subunits image of the Spermatoproteasome in the same column as the standard α-subunit.

Figure 2C is not very informative. There are no labels on the X-axis so what do the heights of each bar mean.

Line 280, change significant to significantly.

Line 356: “Functionally, β5t reduces chymotrypsin-like activity compared to β5 or β5i (Fig. 1E)” – It is unclear what this sentence means. Is it that β5t specificity is not chymotrypsin-like, or is less-chymotrysin-like. If so, please explain further.

Line 467 The subheadings (e.g The 19S regulatory particle) should be highlighted more on page 12 and 13. They blend in with the main text.

Author Response

Comment 1: Figure 1, it would be helpful to label the left boxes as “α-subunits” and the right boxs as “β-subunits”’.

Figure 1F, place the α-subunits image of the Spermatoproteasome in the same column as the standard α-subunit.

Response 1: We thank the reviewer for these comments and have revised the figure accordingly.

Comment 2: Figure 2C is not very informative. There are no labels on the X-axis so what do the heights of each bar mean.

Response 2: We are grateful for the reviewer’s comment. The figure has been revised to improve clarity accordingly.

Comment 3: Line 280, change significant to significantly.

Response 3:  We thank the review for spotting this mistake that has been corrected accordingly.

Comment 4: Line 356: “Functionally, β5t reduces chymotrypsin-like activity compared to β5 or β5i (Fig. 1E)” – It is unclear what this sentence means. Is it that β5t specificity is not chymotrypsin-like, or is less-chymotrysin-like. If so, please explain further.

Response 4: We thank the reviewer for this comment. The text has been revised to improve clarity.

Comment 5: Line 467 The subheadings (e.g The 19S regulatory particle) should be highlighted more on page 12 and 13. They blend in with the main text.

Response 5: We agree with the reviewer’s suggestion and requesting the the editors of Biomolecules to retain the section highlights as they appear in the submitted version of the manuscript.

Reviewer 2 Report

Comments and Suggestions for Authors

In the manuscript, "Splitting Expectations: A Review of the Functional and Catalytic Diversity of the Proteasome," the authors provide a detailed review of the literature on the different forms of proteasomes. The authors provide a detailed description of the structure of proteasomes, the involvement of proteolytic subunits in various regulatory processes, and the potential for the use of proteasome subunit inhibitors. The review is generally interesting and accessible to non experts  of proteolysis. Overall, this is a fairly high-quality review article. There are several recommendations for authors to improve the manuscript:

  1. Section 4, subsection 4.1. Immunoproteasomes.

The immune function of β1i is not limited to the generation of MHC-I-compatible epitopes. These subunits participate in the polarization of M0-M2 macrophages (T. M. Astakhova, N. S. Karpov, N. O. Dashenkova et al. “Inhibition of Proteasome LMP Activity Suppresses Chilz Expression in Mouse Colon Adenocarcinoma Tissue and Restrains Tumor Growth.” Oncol. Res. 2025. 33(9):2573–2595). In addition, immune proteasomes participate in the control of T-lymphocyte expansion [Moebius J. et al. Immunoproteasomes are essential for survival and expansion of T cells in virus-infected mice. Eur. J. Immunol. 2010. Vol. 40, No. 12. P. 3439–3449] and cytokine synthesis [Muchamuel T. et al. A selective inhibitor of the immunoproteasome subunit LMP7 blocks cytokine production and attenuates progression of experimental arthritis. Nat Med. 2009. Vol. 15, No. 7. P. 781–787].

  1. Section 4, subsection 4.2. Intermediate Proteasomes

The manuscript does not contain information on asymmetric 20S proteasomes. Each 20S proteasome contains two rings of beta subunits, thus all subunits are present in pairs. This allows one beta ring to contain a constitutive catalytic subunit and the other an immune subunit. The existence of such 20S proteasomes has been reported in several publications. For example, after cytokine stimulation, 20S proteasomes containing both β5i and β5 were discovered by Freudenburg et al. [Freudenburg W. et al. Reduction in ATP Levels Triggers Immunoproteasome Activation by the 11S (PA28) Regulator during Early Antiviral Response Mediated by IFNβ in Mouse Pancreatic β-Cells. PLoS ONE ed. Mukhopadhyay P. 2013. Vol. 8, No. 2. P. e52408], whereas complexes containing both β1i and β1 were described in the article by Klare N. et al. Intermediate-type 20 S Proteasomes in HeLa Cells: “Asymmetric” Subunit Composition, Diversity and Adaptation. J. Mol Biol. 2007. Vol. 373, No. 1. P. 1–10.

Author Response

Comment 1: Section 4, subsection 4.1. Immunoproteasomes.

Comment 1: Section 4, subsection 4.1. Immunoproteasomes.

The immune function of β1i is not limited to the generation of MHC-I-compatible epitopes. These subunits participate in the polarization of M0-M2 macrophages (T. M. Astakhova, N. S. Karpov, N. O. Dashenkova et al. “Inhibition of Proteasome LMP Activity Suppresses Chilz Expression in Mouse Colon Adenocarcinoma Tissue and Restrains Tumor Growth.” Oncol. Res. 2025. 33(9):2573–2595). In addition, immune proteasomes participate in the control of T-lymphocyte expansion [Moebius J. et al. Immunoproteasomes are essential for survival and expansion of T cells in virus-infected mice. Eur. J. Immunol. 2010. Vol. 40, No. 12. P. 3439–3449] and cytokine synthesis [Muchamuel T. et al. A selective inhibitor of the immunoproteasome subunit LMP7 blocks cytokine production and attenuates progression of experimental arthritis. Nat Med. 2009. Vol. 15, No. 7. P. 781–787].

Response 1: We appreciate the reviewer’s valuable suggestion. The specific studies recommended by the reviewer have been incorporated into the revised version of the manuscript.

Comment 2:  Section 4, subsection 4.2. Intermediate Proteasomes

The manuscript does not contain information on asymmetric 20S proteasomes. Each 20S proteasome contains two rings of beta subunits, thus all subunits are present in pairs. This allows one beta ring to contain a constitutive catalytic subunit and the other an immune subunit. The existence of such 20S proteasomes has been reported in several publications. For example, after cytokine stimulation, 20S proteasomes containing both β5i and β5 were discovered by Freudenburg et al. [Freudenburg W. et al. Reduction in ATP Levels Triggers Immunoproteasome Activation by the 11S (PA28) Regulator during Early Antiviral Response Mediated by IFNβ in Mouse Pancreatic β-Cells. PLoS ONE ed. Mukhopadhyay P. 2013. Vol. 8, No. 2. P. e52408], whereas complexes containing both β1i and β1 were described in the article by Klare N. et al. Intermediate-type 20 S Proteasomes in HeLa Cells: “Asymmetric” Subunit Composition, Diversity and Adaptation. J. Mol Biol. 2007. Vol. 373, No. 1. P. 1–10.

Response 2: We thank the reviewer for this helpful comment. The studies suggested by the reviewer have been included in the revised manuscript.

Reviewer 3 Report

Comments and Suggestions for Authors

In this manuscript, Zachor-Movshovitz et al. review the proteasome's catalytic activities, emphasizing the biochemical and structural features of its β subunits that govern cleavage specificity and their evolutionary diversification into specialized variants such as the immunoproteasome and thymoproteasome. They also discuss how factors like regulatory particles, post-translational modifications, and physiological stressors modulate proteolysis, and highlight the therapeutic potential of selective catalytic-site inhibitors in diseases such as cancer and autoimmunity. Their manuscript is likely to be of interest to the readers of BIOMOLECULES. To further improve the quality of the paper, I recommend addressing the following minor points.

Minor points

  1. Line 354: Is there possibly a double space between 'constitutive' and 'beta5'?
  2. Line 566: Should 'bothe' be corrected to 'both'?
  3. Line 567: Should 'lnked' be corrected to 'linked'?
  4. Line 568: Is there possibly a double space between 'its' and 'intracellular'?
  5. Line 621: Is there possibly a double space between 'constitutive' and 'proteasomes'?
  6. Line 697: The word 'ProPred1' appears to be in bold.
  7. Line 699: The word 'nHLAPred' appears to be in bold.
  8. Line 701: The word 'PCleavage' appears to be in bold.
  9. Line 704: The words 'NetChop has been…' appears to be in bold.
  10. Line 707: The word 'iPCPS' appears to be in bold.
  11. Line 712: The words 'Pepsickle emerged as…' appears to be in bold.
  12. Line 783: The words 'M. tuberculosis' should be italicized.
  13. Line 840: Please remove the leading '1.' at the beginning of Reference 1.
  14. Line 1010: Check References 78 and 79.
  15. Line 1049: References 96 and 97 appear to be the same—please check for duplication.
  16. Line 1105: The journal abbreviations are not consistently formatted. For example, please compare References 120 and 78."
  17. Line: 1151: Drosophila melanogaster should be italicized.

Author Response

Comments: 

Minor points

  1. Line 354: Is there possibly a double space between 'constitutive' and 'beta5'?
  2. Line 566: Should 'bothe' be corrected to 'both'?
  3. Line 567: Should 'lnked' be corrected to 'linked'?
  4. Line 568: Is there possibly a double space between 'its' and 'intracellular'?
  5. Line 621: Is there possibly a double space between 'constitutive' and 'proteasomes'?
  6. Line 697: The word 'ProPred1' appears to be in bold.
  7. Line 699: The word 'nHLAPred' appears to be in bold.
  8. Line 701: The word 'PCleavage' appears to be in bold.
  9. Line 704: The words 'NetChop has been…' appears to be in bold.
  10. Line 707: The word 'iPCPS' appears to be in bold.
  11. Line 712: The words 'Pepsickle emerged as…' appears to be in bold.
  12. Line 783: The words 'M. tuberculosis' should be italicized.
  13. Line 840: Please remove the leading '1.' at the beginning of Reference 1.
  14. Line 1010: Check References 78 and 79.
  15. Line 1049: References 96 and 97 appear to be the same—please check for duplication.
  16. Line 1105: The journal abbreviations are not consistently formatted. For example, please compare References 120 and 78."
  17. Line: 1151: Drosophila melanogastershould be italicized.

Response to all points: We thank the reviewer for identifying these typos and errors, all of which have been corrected in the revised manuscript.